# The Scope for Thalassemia Gene Therapy by Disruption of Aberrant Regulatory Elements

**DOI:** 10.3390/jcm8111959

**Published:** 2019-11-13

**Authors:** Petros Patsali, Claudio Mussolino, Petros Ladas, Argyro Floga, Annita Kolnagou, Soteroula Christou, Maria Sitarou, Michael N. Antoniou, Toni Cathomen, Carsten Werner Lederer, Marina Kleanthous

**Affiliations:** 1Department of Molecular Genetics Thalassemia, The Cyprus Institute of Neurology and Genetics, 2371 Nicosia, Cyprus; petrospa@cing.ac.cy (P.P.); argyrof@cing.ac.cy (A.F.); marinakl@cing.ac.cy (M.K.); 2Institute for Transfusion Medicine and Gene Therapy, Medical Center–University of Freiburg, 79106 Freiburg, Germany; claudio.mussolino@uniklinik-freiburg.de (C.M.); toni.cathomen@uniklinik-freiburg.de (T.C.); 3Center for Chronic Immunodeficiency, Medical Center, Faculty of Medicine, University of Freiburg, 79106 Freiburg, Germany; 4Cyprus School of Molecular Medicine, 2371 Nicosia, Cyprus; 5Thalassemia Clinic Paphos, Paphos General Hospital, 8100 Paphos, Cyprus; an.kolnagou@gmail.com; 6Thalassemia Clinic Nicosia, Archbishop Makarios III Hospital, 1474 Nicosia, Cyprus; chrnchr@spidernet.com.cy; 7Thalassemia Clinic Larnaca, Larnaca General Hospital, 6301 Larnaca, Cyprus; msitarou@yahoo.gr; 8Department of Medical and Molecular Genetics, King’s College London, London SE1 9RT, UK; michael.antoniou@kcl.ac.uk

**Keywords:** CRISPR/Cas9, Cpf1, Cas12a, TALEN, gene editing, gene therapy, thalassemia, advanced therapy medicinal product, ATMP

## Abstract

The common IVSI-110 (G>A) β-thalassemia mutation is a paradigm for intronic disease-causing mutations and their functional repair by non-homologous end joining-mediated disruption. Such mutation-specific repair by disruption of aberrant regulatory elements (DARE) is highly efficient, but to date, no systematic analysis has been performed to evaluate disease-causing mutations as therapeutic targets. Here, DARE was performed in highly characterized erythroid IVSI-110(G>A) transgenic cells and the disruption events were compared with published observations in primary CD34^+^ cells. DARE achieved the functional correction of β-globin expression equally through the removal of causative mutations and through the removal of context sequences, with disruption events and the restriction of indel events close to the cut site closely resembling those seen in primary cells. Correlation of DNA-, RNA-, and protein-level findings then allowed the extrapolation of findings to other mutations by in silico analyses for potential repair based on the clustered regularly interspaced short palindromic repeats (CRISPR)/CRISPR-associated (Cas) 9, Cas12a, and transcription activator-like effector nuclease (TALEN) platforms. The high efficiency of DARE and unexpected freedom of target design render the approach potentially suitable for 14 known thalassemia mutations besides IVSI-110(G>A) and put it forward for several prominent mutations causing other inherited diseases. The application of DARE, therefore, has a wide scope for sustainable personalized advanced therapy medicinal product development for thalassemia and beyond.

## 1. Introduction

Thalassemias are caused by the deficient production of globins. Beta-thalassemia has particular clinical relevance and as a monogenic blood disorder affecting the expression of β-globin (HBB), is an ideal target for therapy development based on gene addition or gene correction. It is also a frequent testbed for more widely applicable advanced therapy medicinal product (ATMP) technology, including the establishment of efficient mutation-specific therapies, which in most cases employ precise sequence repair by homology-directed repair (HDR) or the nascent base or prime editing technologies [1,2]. Despite the early application of zinc finger nucleases (ZFN) to the HBB locus [3,4] and rapid method development for gene editing since, precise correction of mutations is, nevertheless, usually still too inefficient for clinical use [5]. This is equally true for ZFN-based tools, the redesign of which for new targets is notoriously difficult [6], as for the two more recent and more accessible editing platforms, transcription activator-like effector nucleases (TALEN) and clustered regularly interspaced short palindromic repeats (CRISPR)/CRISPR-associated nucleases (Cas). A particular challenge for precise editing by any platform is modification of early stem cells, which are often of greatest therapeutic relevance and in which imprecise repair by non-homologous end-joining (NHEJ) greatly outperforms HDR. Several classes of mutations are potentially amenable to functional repair by effective NHEJ-based strategies. Prominently, NHEJ-based disruption of regulatory elements is able to activate γ-globin as a universal positive disease modifier of β-hemoglobinopathies [7,8,9,10,11,12]. This approach is currently in early clinical trials based on CRISPR (ClinicalTrials.gov Identifier: NCT03655678) and ZFN (ClinicalTrials.gov Identifier: NCT03432364) technologies, and its performance in comparison with other approaches is as yet unclear. Additionally, several mutation-specific strategies based on single NHEJ-based disruption events have already shown promise for other diseases, such as removal of the pathological trinucleotide repeat expansion of the *DMPK* gene for myotonic dystrophy type 1 [13] and disruption of the normal splice acceptor site (SA) to achieve skipping of defective *DMD* exons for Duchenne muscular dystrophy [14]. Importantly, two independent groups have recently shown repair of defective splicing in β-thalassemia, our own for the *HBB^IVSI-110(G>A)^* mutation using CRISPR/Cas9 and TALEN tools [15,16], and Xu et al. for the *HBB^IVSI-110(G>A)^* and *HBB^IVS2-654(C>T)^* mutations using CRISPR/Cas9 and Cas12a (also known as CRISPR from Prevotella and Francisella 1, Cpf1), respectively [17]. Conceptually, NHEJ-based mutation-specific repair by disruption of aberrant regulatory elements (DARE) is suitable for several classes of targets, such as aberrant splice donor (aSD) or acceptor (aSA) sites, cryptic splice sites and the mutations activating those cryptic splice sites. Although DARE is limited by platform-specific sequence requirements [18] and by the need to avoid disruption of coding or other conserved sequences, it is potentially applicable to a vast number of genetic diseases. This is particularly true if disruption of context sequences is sufficient for functional correction of aSD or aSA sites or for the deactivation of cryptic splice sites, all of which would make DARE suitable even where the primary mutation or splice site itself cannot be targeted. We, therefore, set out to perform clonal analyses for a direct correlation of disruption events and functional correction, based on *HBB^IVSI-110(G>A)^*-specific CRISPR/Cas9 and TALEN nucleases validated in primary cells [15,16], and to extrapolate our findings to shortlist other disease-causing mutations potentially suitable for correction by DARE. Our study vindicated the DARE approach for application to many other disease targets, by highlighting an unexpected degree of flexibility for the functional correction of splicing.

## 2. Materials and Methods

### 2.1. Murine Erythroid Leukemia (MEL) HBB^IVS^ Cells

Humanized transgenic murine erythroleukemia cell (MEL) lines holding the mutant, *HBB^IVSI-110(G>A)^*, and normal, *HBB*, human *HBB* gene at an average vector copy number (VCN)/haploid genome ≈2 (VCN), were used for genome-disruption experiments in bulk populations and as normal controls, respectively. In addition, a MEL-*HBB^IVS^* clonal cell line (VCN = 1) was used for the selection and comparison of edited clones. All humanized transgenic MEL cell lines were characterized in a previous study [19]. *HBB^IVSI-110(G>A)^*- and CCR5-specific designer nucleases were designed and validated as described elsewhere [16,20], and delivered as plasmids by electroporation using a Gene Pulser II and 0.4 cm MicroPulser electroporation cuvettes (Bio-Rad Laboratories, Hercules, CA, USA) in Roswell Park Memorial Institute (RPMI) 1640 medium (Life Technologies, Grand Island, NY, USA) with optimized settings (500 μL Vf/400 mV/850 μF/Infinite Resistance; Appendix A). Per sample, 10^7^ cells were treated with 20 μg pEGFP-N1 (Clontech, Palo Alto, CA, USA) reporter and 40 μg nuclease-specific plasmid mix (1:1 molar ratios for TALEN pairs, 1:3 molar ratio for Cas9- and gRNA-encoding plasmids, with the addition of a pUC118 plasmid to normalize the total plasmid amount). MEL cells were harvested in aliquots for flow-cytometry assessment of transfection efficiency and for gDNA-based quantification of targeted disruption by the T7E1 assay [20,21] and TIDE (Tracking of Indels by Decomposition; https://tide.nki.nl/) [22], before being induced to differentiate with 1.5% dimethyl sulfoxide (DMSO) (Sigma-Aldrich, Irvine, UK) and harvested for RNA (day 3) and protein (day 6) extraction. NHEJ-induced indels for TALEN R1/L1, R1/L2, and the RNA-guided nuclease (RGN) were characterized by polymerase chain reaction (PCR) amplification encompassing the target site, bacterial cloning, and sequencing for a total of 100 colonies per designer nuclease. Additionally, NHEJ-induced indels and *HBB* expression were correlated by the limiting dilution and expansion of 96 putative MEL-*HBB^IVS^* cell clones (VCN 1) per nuclease, followed by analyses of expansion-phase gDNA, and differentiation-phase RNA and protein lysate.

### 2.2. Indel Characterization in Transgenic Humanized MEL-HBB^IVS^ Cell Lines

#### 2.2.1. Analysis of Indels in Bulk Cells

For the characterization of indels produced by the NHEJ repair pathway after treatment of MEL-*HBB^IVS^* cell pools with specific designer nucleases and the nuclease-free negative pUC118 control, PCR products encompassing exon 1, intron 1, and part of exon 2 were cloned into pCR.4 Blunt-TOPO^®^ vectors using the Zero Blunt^®^ TOPO^®^ PCR Cloning Kit and TOP10 chemically competent bacteria (Invitrogen, Thermo Fisher Scientific, Carlsbad, CA, USA), according to the manufacturer’s instructions. A total of 100 colonies each were picked for sequencing across the cleavage site in order to characterize indels produced by specific designer nucleases, TALEN R1/L1, R1/L2, and RGN, and ≈30 colonies from nuclease-free negative controls, pUC118. The alignment of the sequencing traces was performed using the SnapGene software (GSL Biotech, Chicago, IL, USA, available at www.snapgene.com).

#### 2.2.2. Analysis of Indels in Disrupted MEL-*HBB^IVS^* Clones

MEL-*HBB^IVS^* cell clones (VCN 1) were selected from TALEN- and RGN-edited bulk populations via limiting dilution in 96-well plates. Plates were incubated for 48 h before they were scored microscopically for single colonies per well. Wells with single clones were expanded to 24-well plates in order to allow gDNA extraction and functional analyses. All isolated clones were cryopreserved until genome-disrupted clones were characterized by two rounds of Sanger sequencing (pre- and post-cryopreservation). Clones were induced to differentiate in parallel with the non-edited controls (two untransfected controls (UT) and two pUC118) in RPMI 1640 supplemented with 1x penicillin/streptomycin, 10% FBS, 1.5% DMSO (all Invitrogen, Thermo Fisher Scientific, Carlsbad, CA, USA) for six days. Induced cells (5 mL) were collected on day 3 and 6 for RNA extraction (5 × 106 cells/1 mL TRIZOL™ (Invitrogen, Thermo Fisher Scientific, Waltham, MA, USA)) and on day 6 for protein extraction (5 × 106 cells/50 μL radioimmunoprecipitation assay (RIPA) lysis buffer).

### 2.3. Sequencing

Purified plasmids and PCR products were sequenced using the BigDye Terminator v1.1 Cycler sequencing kit (Applied Biosystems, Foster City, CA, USA), including 1 × GC-RICH solution (Roche, Basel, Switzerland). DNA sequencing products were purified using Performa^®^ DTR Gel Filtration Cartridges Performa^®^ DTR Gel Filtration Cartridges (Edge Biosystems, Gaithersburg, MD, USA, USA) and analyzed on a Hitachi 3031xl Genetic Analyzer with the Sequence Detection Software version 5.2 (Applied Biosystems, Foster City, MA, USA).

### 2.4. Assessment of RNA Expression

RNA was isolated from transgenic MEL cells using TRIZOL™ and treated with DNase I (Invitrogen, Thermo Fisher Scientific). After reverse transcription using the TaqMan Reverse transcription PCR kit and random hexamers (Applied Biosystems, Foster City, CA, USA) to produce cDNA as template for qPCR measurements, variant-specific quantification of samples was performed by probe-based duplex PCR with the Multiplex PCR Kit (Qiagen, Hilden, Germany), in triplicate against a plasmid-based standard curve holding the aberrant and normal amplicons, as published elsewhere [23]. By contrast, total human HBB expression was quantified by SYBR Green-based measurement (Applied Biosystems, Foster City, CA, USA). Variant-specific and total HBB measurements were normalized with Hba1/2 as a calibrator for variation in erythroid differentiation using the 2^−Δ(ΔCt)^ analysis method. Non-edited samples were used as negative controls. Resulting absolute quantities for each variant mRNA were used to assess the percentage of each mRNA variant in the total population, as well as its percentage difference relative to controls. The sequences of primers and probes (Metabion International AG, Martinsried, Germany) were as follows, for SYBR Green-based assessment of total HBB expression (hHBB_EX1_FW_1: GGTGCATCTGACTCCTGAG/hHBB_EX1_RV_2_A: CACCACCAACTTCATCCAC and hHBB_FW_EX2_B: GGCAAGAAAGTGCTCGG/hHBB_EX2.3_RV_B: GTGCAGCTCACTCAGTG) with murine actin (mouse β-Actin FW: GCTTCTTTGCAGCTCCTTCGT/mouse β-Actin RV: CCAGCGCAGCGATATCG) and Hba-1/a2 (mHba-a1/a2 FW: GTCACGGCAAGAAGGTCGC/mHba-a1/a2 RV: GGGGTGAAATCGGCAGGGT) [24] as calibrators, and for probe-based assessment of HBB mRNA variants (hHBB_EX1_FW_3: GGGCAAGGTGAACGTG/hHBB_EX2_RV_1: GGACAGATCCCCAAAGGAC with probes wtHBB_Probe_ZNA: 6-FAM-TGGG(PDC)AGG(PDC)TG(PDC)TG-ZNA-3-BHQ-1 and IVSI-110_MGB_Probe: VIC-TAAGGGTGGGAAAATAGA-MGB).

### 2.5. Globin Chain Analyses

Immunoblots were performed according to standard protocols, as described previously [19]. Protein lysates from 0.5–1 × 10^6^ humanized MEL cells on day 6 of the DMSO-induced end-point differentiation were separated by polyacrylamide gel electrophoresis and blotted onto nitrocellulose Parablot NCP membranes (Macherey–Nagel GmbH, Düren, Germany) using wet electrophoretic transfer. Membranes were temporarily stained with Ponceau S solution (Sigma-Aldrich, St. Louis, MO, USA) to confirm the successful transfer of the proteins, before they were blocked in blocking buffer (1% BSA (Sigma-Aldrich, St. Louis, MO, USA) in TBS-Tween) and incubated with the appropriate primary antibody, specifically Mouse-αHuman HBB (@1:1000; clone 37-8; #sc-21757), Rabbit-αMouse Hba (@1:1000; H80; #sc-21005) (all Santa Cruz Biotechnologies, Dallas, TX, USA), or Mouse-αMouse-Actb (@1:10000; clone AC-15; #A1978; A1978, Sigma-Aldrich, Dorset, UK). Membranes were washed and incubated with the appropriate horseradish-peroxidase-conjugated secondary antibody, specifically Goat-αMouse-Immunoglobulin (Ig) G(H+L) (#115-035-003) or Goat-αRabbit-IgG(H+L) (#111-035-003) (both @1:10000; Jackson ImmunoResearch Laboratories, Suffolk, UK). Protein bands were visualized and quantified using ImageLab Software 5.1 (Bio-Rad Laboratories, Hercules, CA, USA). Murine Actb was used as a protein loading control and murine Hba in order to normalize for erythroid differentiation. Correction of the *HBB^IVSI-110(G>A)^* mutation is reported here as the fold-change of the ratio of the HBB/Hba chain band intensities in the edited samples relative to the negative control and as the percentage relative to the MEL-*HBB* positive control.

### 2.6. Statistical and Bioinformatics Analyses

#### 2.6.1. Statistical Analyses

Data transformations were performed in Office 2016 Excel (Microsoft, Inc., Santa Rosa, CA, USA). Spearman correlation analyses and group-wise comparisons were conducted using Prism 8.0 (GraphPad Software Inc., San Diego, CA, USA). Samples were tested for normality by Shapiro–Wilk test, and group-wise comparison was performed by parametric analyses (one-way ANOVA with Dunnett’s post-hoc test, where all samples within the detection limit passed the normality test) and a non-parametric test (Kruskal–Wallis with Dunn’s post-hoc test, where at least one sample failed the normality test), as appropriate.

#### 2.6.2. Retrieval of Target Sequences

*HBB* mutations were retrieved as a global list from the ITHANET Portal (www.ithanet.eu) [25] and filtered in Excel for base substitutions or short (<5 nt) sequence additions affecting *HBB* outside the coding sequence, with discernible β-thalassemia carrier or patient phenotype (Appendix A). Mutations not characterized as pure loss-of-function mutations of critical positive regulatory elements of transcription and falling outside the characterized regulatory sequences of the distal CACCC box (−108 to −100 nt from the CAP site), the proximal CACCC box (−93 to −85 nt), the CCAAT box (−76 to −72), the TATA box (−31 to −26) [26] and the conserved splice consensus sites GT…AG of introns 1 and 2, were included in the analyses for potential designer nucleases. Based on reference sequence NG_059281.3 representing the human *HBB* gene and retrieved from ENSEMBL [27], mutation-specific sequences containing individual single-nucleotide polymorphisms (SNPs) were created in Vector NTI 11.5 and used in platform-specific in silico design tools.

#### 2.6.3. In Silico Design and Evaluation of Designer Nucleases

In silico design and evaluation of Cas9 and Cas12a guide RNAs (gRNAs) targeting aberrant or cryptic splice sites was performed using the CRISPOR webtool (http://crispor.tefor.net/) [28] based on mutant HBB sequences and the presence of the appropriate protospacer adjacent motif (PAM) for Cas9 (20-nt gRNA 3′ -NGG PAM) and Cas12a (5′ TTTV- PAM and 23-nt gRNA), respectively. Doench ‘16 Score for Cas9 and DeepCpf1 score for Cas12a were employed for efficiency predictions on a scale 0–100, with 100 being the highest efficiency for the specific position. Off-targeting analysis of Cas9 gRNAs (MIT specificity score) was based on a genome database preset Homo sapiens—Human—UCSC Feb. 2009 (GRCh37/hg19) + SNPs: 1000 Genomes and ExaC on a scale 0–100, with 100 being the highest specificity.

In silico design and evaluation of TALEN pairs was performed using TAL Effector Nucleotide Targeter 2.0 (https://tale-nt.cac.cornell.edu/) [29]. Specific design conditions included spacer length 10–15, repeat array length 15–20, upstream base: T only, G substitute: NH, filter option: hide redundant TALENs, pre-loaded sequence: human genome, Scoring matrix: Doyle et al. and deselection of Streubel et al. guidelines, followed by shortlisting of TALEN pairs predicted to cleave outside exons and closest to the target sites, based on the best possible RVD score calculated by the TALE-NT 2.0 Paired Target Finder.

## 3. Results

### 3.1. Analysis of DARE in Bulk Populations of Modified Polyclonal MEL-HBB^IVS^ Cells

In order to allow the clonal analysis of the effect of indel events on *HBB^IVSI-110(G>A)^*-derived expression from a single locus and without the interference of endogenous HBB-derived expression, we drew on the humanized transgenic polyclonal MEL-*HBB^IVS^* cell line published elsewhere [19]. In brief, polyclonal MEL-*HBB^IVS^* cells contain stably integrated copies (VCN = 1.9) of the mutated GLOBE *HBB^IVSI-110(G>A)^* LV transgene and are characterized by 45% correctly and 55% aberrantly spliced human HBB mRNA, similar to ratios observed in cultured patient-derived HSPCs. At the protein level, MEL-*HBB^IVS^* cells express ≈5% of functional globin chain levels observed in their normal counterpart MEL-*HBB* (VCN ≈ 2), similar to the ratio seen in peripheral blood of *HBB^IVSI-110(G>A)^* homozygous patients compared to normal controls [19].

Owing to the reported difficulties in designing ZFN reliably, we restricted all analyses to TALEN and RGN platforms. Designer nucleases of either platform were delivered to MEL-*HBB^IVS^* cell lines by conventional plasmid electroporation at optimized conditions (400 mV, 1050 μF), which allowed ≈65% transfection efficiency at ≈20% cell death (Appendix A). A triplicate repeat in MEL-*HBB^IVS^* at (54.5 ± 11.2)% average transfection efficiency showed similar targeted disruption efficiencies of the mutant *HBB^IVSI-110(G>A)^* transgene for both TALEN pairs, (28.0 ± 17.0)% for R1/L1 and (26.8 ± 20.9)% for R1/L2, whereas efficiencies for the RGN remained low (5.3 ± 2.2)% (Appendix A). For the most efficient experiment with these three nucleases, in which 51.7%, 56.4%, and 8.35% targeting efficiencies, respectively, were obtained for the *HBB^IVSI-110(G>A)^* transgene according to the T7E1 assay (Figure 1a), we additionally employed tracking of indels by decomposition (TIDE) [22] for independent assessment of targeted disruption and in order to characterize the population of edited alleles for the proportion of different indel sizes (Figure 1b). The TALENs under study predominantly produced deletions rather than insertions, in line with previous reports for TALENs [30], and TALEN R1/L1, with its 13-bp spacer, gave deletions of 4–8 bp, whereas TALEN R1/L2, with its shorter, 10-bp spacer, gave predominantly shorter deletions of 1–9 bp. For the same experiment, allele populations for each nuclease were additionally characterized by sequencing 100 individual bacterial clones of PCR amplicons covering the *HBB^IVSI-110(G>A)^* region. This gave 41%, 38%, and 5% edited clones for TALEN R1/L1, TALEN R1/L2, and RGN, respectively (Appendix A), whereas the size of deletions for TALEN R1/L1 (Range 2–54 bp, Median 8 bp, Mode 16 bp) and TALEN R1/L2 (Range 1–64 bp, Median 7 bp, Mode 4 bp) confirmed the differential size distributions detected by TIDE. Modifications by both TALENs had a similar distance from the normal SA site (Median 20 bp, Mode 22 bp for both). Importantly, for both TALENs the majority of indels occurred upstream of the *HBB^IVSI-110(G>A)^* mutation. For TALEN R1/L1, the AG motif of the aberrant SA site was disrupted in only 16 out of 41 edited bacterial clones, one of which had also lost its normal SA site. A similar pattern was observed for TALEN R1/L2, where the aberrant SA site was disrupted in 13 out of 38 edited bacterial clones, one of which extended into exon 2. Finally, in the case of RGN-mediated disruption, only five clones out of 100 showed modification events, four of which disrupted the aberrant SA site. Overall, for each of the three designer nucleases, the number of edited clones was in line with expectations from T7E1 assays (see Figure 1a).

The degree of functional correction after treatment was assessed at the RNA and protein level on days 3 and 6 of induced differentiation, respectively, in genome-edited MEL-*HBB^IVS^* bulk populations compared with all-negative (no-electroporation non-edited) control. At the RNA level, both TALEN pairs, R1/L1 and R1/L2, showed a significantly increased ratio of correct/aberrant HBB transcripts, 2.39 and 2.2, respectively, compared with 0.81 for the control (Figure 1c). By contrast, the ratio of correct/aberrant HBB transcript ratios in the RGN-edited bulk population (0.84) remained close to the overall range of the all-negative control, nuclease-negative controls, and nucleases targeting *CCR5* as the control target (0.48–0.81). At the protein level, a similar pattern of functional correction was observed, and the HBB/Hba ratio detectable in immunoblots (Figure 1d) was significantly increased across multiple experiments up to 0.76 and 0.83 in the TALEN R1/L1- and R1/L2- edited populations, respectively, compared with 0.36 detected in the all-negative control (Figure 1e). A non-significant increase (0.48) was observed in RGN-edited bulk populations.

Taken together, electroporation of plasmids encoding our *HBB^IVSI-110(G>A)^-specific* nucleases in MEL-*HBB^IVS^* cells gave superior results for TALENs compared with the RGN, in contrast to our previous findings for delivery as mRNAs or as ribonucleoprotein complexes [16]. More importantly, even plasmid-based delivery achieved significant functional correction of MEL-*HBB^IVS^* at the transcriptional and translational level in edited bulk populations without enrichment, thus vindicating DARE as a robust therapeutic approach.

### 3.2. Analysis of DARE in Cell Clones of Modified Clonal MEL-HBB^IVS^ Cells

In order to correlate indel events and functional correction directly, we then analyzed DARE in clonal MEL-*HBB^IVS^* cells with a single *HBB^IVSI-110(G>A)^* copy per cell [19](Figure 2). The limiting dilution post-treatment and sequencing of 96 potential clones per designer nuclease allowed us to identify and proceed with 34 cell clones that represented individual indel events (Figure 2, gDNA panel), in addition to four clones without modification as negative controls (clone A1#1 after TALEN R1/L1, B1#1 after TALEN R1/L2 and Mock A and B after pUC118 plasmid electroporation). In line with general disruption efficiencies after plasmid-based delivery, TALEN R1/L1 was represented by 14 clones, TALEN R1/L2 by 20 clones, and RGN by 0 clones. After expansion and induced erythroid differentiation of individual clones, the material was collected for DNA-based confirmation of clone identity and purity, for RNA-based assessment of normal *HBB* transcript levels and normal:aberrant transcript ratios, and for the quantification of HBB protein levels.

At the RNA level and based on RT-qPCR (Figure 2, *RNA* panel), restoration of splicing was observed in all clones for which the *HBB^IVSI-110(G>A)^* mutation had been disrupted, with the exception of clones B1#20 and B1#21, in which large deletions also disrupted the normal SA site. Correct splicing was also restored by many indel events that disrupted the region upstream of the *HBB^IVSI-110(G>A)^* mutation but left the aberrant SA site intact. In general, we noticed that most deletions larger than 5 bp (starting from B1#8) completely corrected *HBB* pre-mRNA splicing. Notable was the unique case of complete restoration of correct splicing in clone A1#2 with a single-base (T) deletion. In total, 23 out of 34 genome-edited MEL-*HBB^IVS^* clones had an increase of normal *HBB* mRNA by at least 40%. However, deactivation of aberrant mRNA alone appears to have a therapeutic effect for β-thalassemia, as shown elsewhere [19], so that DARE-mediated depletion of aberrant mRNA may indicate therapeutic potential even more than an increase in normal mRNA. In this vein, 22 out of 34 genome-edited clones had virtually abolished incorrect splicing with ≤ 1.1% aberrant mRNA, at an average of (0.13 ± 0.32)%.

At the protein level and based on immunoblots (Figure 2, *Protein* panel), the correlation of differentiation-normalized HBB levels (HBB/Hba) with normal HBB mRNA levels was moderate to strong (r^2^ = 0.4). The 32 modified clones with an intact *HBB* open reading frame all showed increased HBB expression (by 19.3 ± 11.0 fold, compared to unmodified controls). Predictably, for clones with large deletions (B1#20 and B1#21) extending into *HBB* exon 2, HBB production was close to the background (0.3 and 0.2 of HBB seen in unmodified controls), despite good levels of differentiation (0.86 and 0.63 of Hba seen in unmodified controls). In addition to the increased HBB levels, most modified clones also had lowered levels of differentiation (0.68 ± 0.43 fold, compared to unmodified controls), which in this non-thalassemic model might have been brought about by toxicity of excessive combined murine and human β-globin expression. In this context, it is of note that clones A1#5, A#6, B1#6, and B1#7 carried the same indel and showed similarly high residual aberrant HBB mRNA percentages at low total HBB mRNA levels, at variably low HBB protein expression and at highly variable differentiation. It, therefore, appeared that the four-nucleotide deletions in question marginally reduced aberrant splicing, resulting in unpredictable variability of erythroid differentiation, to which random changes during clonal expansion from single cells might further have contributed. Taken together, these data establish for the first time that disruption of the upstream region of *HBB^IVSI-110(G>A)^* is sufficient to restore splicing in the presence of the primary mutation, highlighting the importance of the flanking of regulatory sequences and of the distance from the branchpoint site (BPS) in the selection of the aberrant vs. the normal SA site.

Besides comparing HBB protein expression after correction with that found in unmodified MEL-*HBB^IVS^* cells, we also compared it to the HBB expression found in polyclonal MEL-*HBB* cells, the reference for normal HBB expression in this model. Clonal MEL-*HBB^IVS^* cells (VCN = 1) express ≈4.2% (1/24) of the differentiation-normalized HBB protein levels found in polyclonal MEL-*HBB* cells (VCN = 2; (100.0 ± 22.2)%) on day 6 of erythroid differentiation, as has been shown elsewhere [19]. Saturation effects of high-level gene expression make normalization for differential gene dosage difficult, so that we disregarded the lower copy number in clonal MEL-*HBB^IVS^* cells and conservatively assumed a 24-fold increase as representing the full restoration of functional HBB expression (Figure 2 Protein panel). Of the 34 clones analyzed, a total of 16 clones (10 edited by TALEN R1/L1 and six by TALEN R1/L2), reached or exceeded normal HBB levels, six further clones exceeded 50% of normal expression, and another six clones exceeded 30% of normal expression. Accordingly, 16/34 clones would have been completely therapeutic and 28/34 clones would have given substantial therapeutic benefit if these fold-changes in HBB expression had been observed in *HBB^IVSI-110(G>A)^*-homozygous CD34^+^ cells.

The observed distribution of indels tied in with deletion patterns in bulk MEL-*HBB^IVS^* cultures (Appendix A) and with target alignments of both nucleases on *HBB^IVSI-110(G>A)^* [16]. Moreover, the indel patterns observed in bulk and clonal analyses significantly correlated with those observed in CD34^+^ cells [16] for deletion size (*p* = 0.007) and for maximum distance of deletions from the expected cleavage site (*p* = 0.005) (Appendix A), emphasizing the relevance of the MEL-*HBB^IVS^* data for the prediction of on-target action and functional correction. In this context, clonal analyses in MEL-*HBB^IVS^* provide direct evidence for the suggestion from our work in primary cells that deletion of context sequences of the aberrant splice site, or the reduction of its distance to the BPS, would be sufficient to achieve functional correction.

### 3.3. Analysis of Potential Alternative DARE Targets for β-Thalassemia

Our findings, therefore, suggested that DARE-based strategies would be suitable at high bulk efficiencies for any target site at least 5 nucleotides from coding or otherwise conserved sequences, following the strategy indicated in Appendix A. For aberrant or cryptic splice sites, in particular, nuclease design might be offset by several bases from the site itself and still have significant therapeutic effect. We, therefore, proceeded to identify potential targets for disruption amongst all know β-thalassemia mutations [25] that fulfilled those position requirements and for which disruption would conceivably have at least partial therapeutic effect (Appendix A). Of those, mutations with a detectable contribution to disease pathology, i.e., with the absence of HΒΒ expression (β^0^), significant reduction of HΒΒ expression (β^+^) or with an established contribution to at least thalassemia intermedia in compound heterozygote conditions, were analyzed further. Finally, those mutations with a mixed mechanism of disease causation, i.e., combining partial loss of function (LoF) with activation of cryptic splice sites, those creating new aberrant splice sites, as well as any non-exonic cryptic splice sites were shortlisted further as the most promising targets for DARE-based therapy of β-thalassemia (Table 1).

Notably, the systematically shortlisted targets included both mutations already successfully addressed by DARE in primary hematopoietic stem and progenitor cells [16,17,46], specifically IthaID 113 (IVSI-110 G>A) and IthaID 211 (IVSII-654 C>T). Application of cutting-edge bioinformatics tools for the design of artificial nucleases [28,29] under consideration of platform-specific design criteria, such as the requirement of a suitable PAM sequence for RGNs, revealed the suitability of all of the shortlisted target sequences for functional assessment and potential therapy by DARE, based on two or more of the three commonly used nuclease platforms *Streptococcus pyogenes* CRISPR/Cas9 (PAM -NGG), Cas12a (conventional PAM: TTTV-), and TALEN (Figure 3).

### 3.4. Additional Disease Targets for DARE

Literature and database searches based on our findings revealed that a great number of mutations and diseases could be repaired by DARE. While most individual genetic diseases and in particular patients stratified for specific underlying mutations are rare, several potential DARE targets outside the field of hemoglobinopathies have a high relative carrier frequency and would, therefore, be of interest for clinical translation and commercial exploitation, with examples covering major organ systems shown in Table 2.

Splicing mutations for many disorders have been tackled successfully, but with mostly subtherapeutic efficiencies, with antisense oligonucleotide approaches, for instance, for Leber congenital amaurosis [50,51,52], Stargardt disease [54], choroideremia [54,65], Miyoshi myopathy [55], and erythropoietic protoporphyria [57]. Similar to what has been achieved for *HBB^IVSI-110(G>A)^* thalassemia by moving from the antisense oligonucleotide application [66] to DARE [16], those same mutations and many more may instead be addressed by potentially highly efficient, disruption-based curative therapies. Of note, the CFTR mutation listed as rs397508266, together with two other splice site mutations, has already been targeted successfully by a dual-nuclease approach for excision [60]. Our data indicate that for this and the other significant mutations listed in Table 1, the single-nuclease approach presented here would be sufficient for restoration of normal splicing at the reduced potential for off-target activity, and at comparable or better on-target efficiency compared to excision based on a pair of nucleases.

## 4. Discussion

In this study we present clonal data for single-nuclease genome disruption of aberrant regulatory elements (DARE) and functional restoration of aberrant splicing in *HBB^IVSI-110(G>A)^* β-thalassemia at the DNA, RNA, and protein level, show significant correlation of findings in the utilized MEL-*HBB^IVS^* cell model with our published data in CD34^+^ cells, and extrapolate our findings to additional β-thalassemia mutations and prominent mutations in other genetic diseases. For β-thalassemia, we suggest suitable designer nucleases based on the Cas9, Cas12a, and TALEN platforms for successful application of DARE.

In the clonal analysis, we found that most events removing upwards of 5 nt from the target sequence and in particular also indels leaving the primary mutation intact had a therapeutic effect for *HBB^IVSI-110(G>A)^* thalassemia. This provides flexibility in target design and renders DARE a therapeutic approach of potentially wide-ranging applicability. Suitable for the removal of any gain-of-function regulatory element at several nucleotides distance from open reading frames or other conserved elements, DARE might be most reliably applied to aSA or aSD sites or inadvertently activated cryptic splice sites, owing to our detailed understanding of splicing regulation.

Resources for therapy development are preferentially invested in severe and common mutations so that several of the mutations, listed in Table 1 will have low priority for commercial therapy development. For instance, specific correction of a mild mutation might prove insufficient for significant correction of disease parameters in compound heterozygous conditions with more severe mutations. Similarly, several of the β-thalassemia targets selected for nuclease design have low frequency and may, therefore, not see the development of mutation-specific therapies in practice. However, target sites with potential impact on several mutations, such as the cryptic splice site IthaID3445 or a cleavage site upstream of IthaID107 (IVS I-5 G>A) and IthaID111 (IVS I-6 T>C), which would affect both mutations, would be of practical interest. Likewise, the *HBB^IVSI-110(G>A)^*-specific nucleases employed in the present study may also prove therapeutic for patients carrying the mutation IthaID114 (IVS I-116T>G) [35]. In this context, it is important to note that the effect of DARE in bulk population in the vicinity of splice consensus sites, branchpoint, and context sequences will be a mixture of activating and deactivating events for the normal, aberrant or cryptic splice sites concerned. In particular, for IthaID107 and IthaID111 close to the intron 1 splice donor of *HBB*, a large proportion of indel events may turn out to be non-therapeutic, but this can only be established by experimental evidence.

For several mutations identified as potential therapeutic targets, the exact mechanism of disease causation is yet to be established. For instance, would the removal of cryptic splice sites at positions IthaID3445 and IthaID3446 favor the use of the normal splice site, or would splicing switch to yet another cryptic site instead? Likewise, the effect of IthaID215-217 and IthaID2183 on splicing of *HBB* intron 2 has not been validated experimentally, so that the effect of their removal by disruption is equally unclear. In all these instances, the designer nucleases put forward by this study would be tools for functional analysis, in addition to their being potential tools for therapy.

Literature searches and existing mutation databases dedicated to splicing readily reveal that there are mutations suitable for this approach in over 180 genes, responsible for many human diseases in which DARE could, therefore, have essential therapeutic effects [67,68]. In this context it is important to note that for some targets or applications ZFN-based tools, though not covered by our analyses, may be a superior choice, because size and repetitive sequences of TALEN, and size and PAM restriction of RGN may favor the ZFN platform, once an efficient and specific nuclease has been established [69].

Whatever platform is used, our observations inform future selection of target sites for NHEJ-based strategies. Investigation of indel patterns in both, cell lines and primary cells, and our clonal analyses and direct functional correlation of indels in MEL-*HBB^IVS^* cells showed that DARE-based functional correction is suitable even for treatment of exon-proximal mutations. Further inferences can be drawn for DARE-specific target site selection and nuclease design for effective and safe functional correction. The observed distribution of indels reported here tied in with target alignments of both nucleases on *HBB^IVSI-110(G>A)^*, and our published TIDE analyses and targeted deep-sequencing results in CD34^+^ cells [16]. Additionally, our present findings in MEL-*HBB^IVS^* cells give direct evidence that disruption of the *HBB^IVSI-110(G>A)^* upstream region is sufficient to restore splicing and protein expression completely, highlighting the general importance of the flanking regulatory sequences and of the distance from the BPS as parameters for the correction of aberrant splicing by DARE. This reflects the importance of specific regulatory sequence elements, such as the polypyrimidine tract and the BPS, for pre-spliceosome assembly and pre-mRNA processing [70,71], and of the distance between BPS and the aSA. Alteration of the BPS–SA distance might impair selection of the aSA and favor correct splicing [72], and in this study might have been at the heart of full splice correction for deletion events ≥5 bp. Targeting of context sequences by DARE would, thus be a promising alternative strategy for functional correction in cases where the primary mutation is not a suitable target for disruption, for instance, because of adjacent essential sequences or because of an absence of suitable PAM sites for currently available RGNs [73].

## 5. Conclusions

This study has shown that DARE represents a flexible and highly efficient therapeutic option for genetic diseases where faithful repair is not required. Accelerated discovery of primary mutations in non-conserved regions, ongoing clinical validation of disruption for other therapeutic strategies, and increasingly favorable conditions for mutation-specific curative therapies through technological, regulatory, and market developments [1], all point to a likely future clinical translation of curative therapies based on DARE for thalassemia and other diseases.

## Figures and Tables

**Figure 1 jcm-08-01959-f001:**
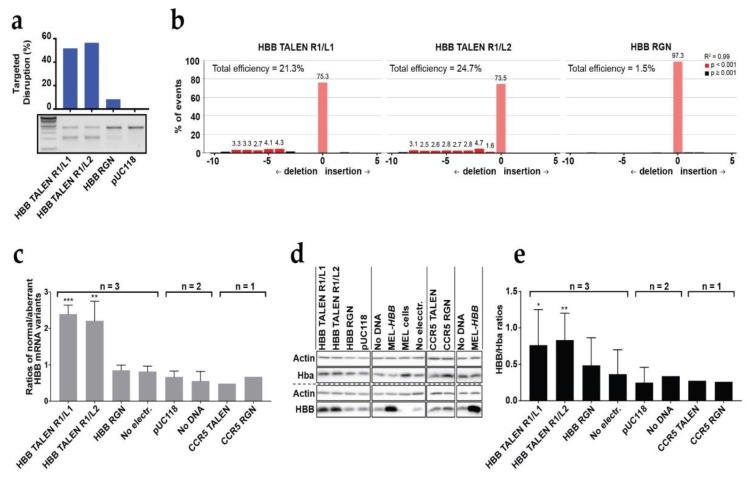
DARE-based correction in humanized transgenic MEL-*HBB^IVS^* bulk cell populations. *HBB*-specific designer nucleases were applied in comparison to *CCR5*-specific designer nucleases and the *GLOBE* gene addition vector as a negative and positive reference, respectively, for HBB induction. (**a**) The T7 endonuclease I assay depicting *HBB^IVSI-110(G>A)^*-targeted disruption in TALEN- (R1/L1 or R1/L2) and RGN-edited cells relative to nuclease-free, pUC118-transfected negative controls (pUC118). (**b**) TIDE analysis depicting *HBB^IVSI-110(G>A)^*-targeted disruption efficiencies of TALEN- (R1/L1 or R1/L2) and RGN-edited cells relative to pUC118. Same-size indels events (-10 deletions to +10 insertions) are scored as a percentage of the total number of events. Significance cutoff was the TIDE default (*p* value < 0.0001). (**c**) The splice correction is shown at the transcript level as the ratios of normal/aberrant *HBB* mRNA levels (±SD) on day 3 of induced differentiation. Significant group-wise comparisons of ratios to no-electroporation controls for samples analyzed in triplicate by unmatched one-way ANOVA and Dunnett multiple-comparison correction: HBB TALEN R1/L1 *** *p* = 0.0007, R1/L2 ** *p* = 0.0016. (**d**) Representative immunoblot of human HBB, murine α-globin (Hba), and actin as calibrator for equal loading on day 6 of induced differentiation. Different same-blot, same-membrane hybridizations are separated by a hashed line. (**e**) Splice correction shown at the protein level as the ratios of HBB/Hba (±SD) on day 6 of induced differentiation, measured by immunoblots as shown in (**d**). Significant group-wise comparisons of ratios to no-electroporation controls for samples analyzed in triplicate by matched one-way ANOVA, assumed sphericity and Dunnett multiple-comparison correction: HBB TALEN R1/L1 * *p* = 0.0118, HBB TALEN R1/L2 ** *p* = 0.0054.

**Figure 2 jcm-08-01959-f002:**
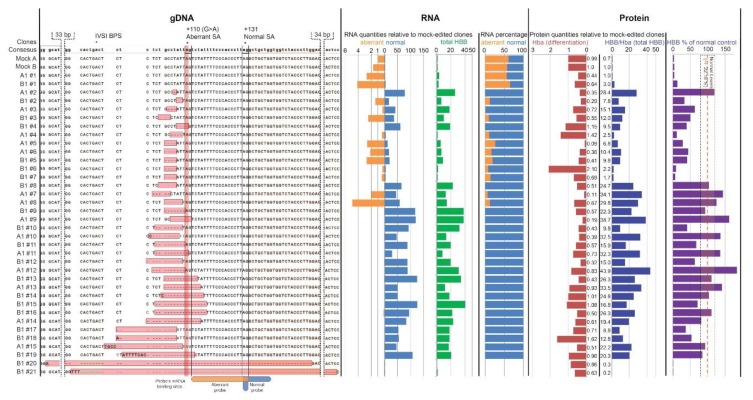
Correlation of indels with restoration of splicing at DNA, transcript, and protein level in MEL-*HBB^IVS^* clones. (gDNA level) Alignment against the *HBB* consensus sequence of the *HBB^IVSI-110^*-proximal regions of MEL-*HBB^IVS^* for 38 MEL clones, specifically two clones unmodified (A1#1 and B1#1) and 34 clones modified after the treatment with TALEN R1/L1 (A1) and R1/L2 (B1), and two mock-edited clones (Mock A and B), showing intron 1 unshaded, the intron-1 branchpoint site (IVSI BPS) in green, exon 2 in orange, the *HBB^IVSI-110(G>A)^* mutation in red, and NHEJ-induced indels in pink. Aberrant (+110 (G>A) Aberrant SA) and normal (+131 Normal SA) splice acceptor sites are underlined (ag) sequences on the consensus sequence. Binding sites for aberrant and normal reverse-transcription quantitative PCR (RT-qPCR) probes are indicated in orange and blue, respectively, below the alignment. (RNA level) The effect of indel events on *HBB* RNA type and quantity was assessed via RT-qPCR of RNA extracts on day 3 of the induced differentiation and is indicated, from left to right, as Hba-normalized expression levels calculated with the ΔΔCt method of the aberrant (orange bars), normal (blue bars), and total (green bars) *HBB* transcripts relative to mock-edited clones (values <1 indicating downregulation), and as the relative percentage of normal and aberrant transcripts as part of the total, calculated with the standard-curve-based quantification of each transcript. (Protein level) The effect of indel events on HBB protein level was assessed via immunoblots on protein extracts on day 6 of induced differentiation and is indicated as the fold difference of the total HBB globin chain levels (HBB/Hba) (blue bars) relative to Mock A. Levels of erythroid differentiation are shown as fold difference of Hba globin chain levels (Hba) (red bars) relative to Mock A and as the percentage of HBB/Hba levels (purple bar) relative to those in the MEL-*HBB* controls. The yellow box indicates the standard deviation (±22.18%) of normal HBB levels seen in induced-differentiated MEL-*HBB* controls (n = 3).

**Figure 3 jcm-08-01959-f003:**
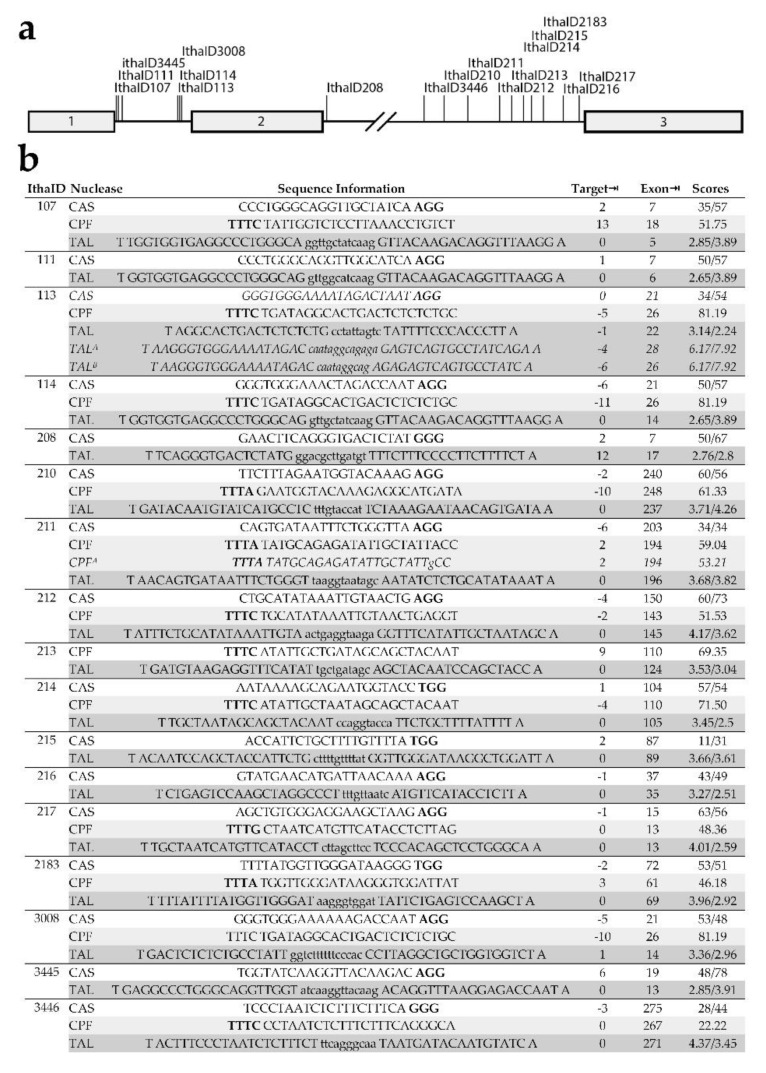
Artificial nucleases for functional assessment and potential therapy by DARE. Shortlisted *HBB* target sites and corresponding Cas9, Cas12a, and TALEN nucleases. (**a**) Schematic representation of *HBB* and the position of the shortlisted target sites. Exons are shown as boxes. (**b**) Artificial nucleases shortlisted for functional assessment and potential therapy by DARE. Design is based on constraints imposed by *CRISPOR* (under default conditions) for Cas9 and Cas12a guide RNAs and by *TALEN Effector Nucleotide Targeter* for TALEN (spacer length: 10–15 nt, RVD length: 15–20, see Experimental Section for further custom parameters). *IthaID* identifies the target by Ithanet ID (www.ithanet.eu) and *Nuclease* the type of nuclease, Cas9 [CAS], Cas12a/Cpf1 [CPF] or TALENs [TAL]. *Sequence Information* in the case of Cas9 and Cas12a present gRNA sequences (5′–3′) with the protospacer adjacent motif (PAM) sequence (-NGG and TTTN-, respectively) shown in bold, whereas in the case of TALENs, upper-case letters indicate the binding sites of TALEN pairs and minor-case letters give the spacer sequence. The preceding T and trailing A for TALEN are not represented by RVDs in TALEN monomers, for which RVD usage is as follows: A = NI, C = HD, G = NH, T = NG, excepting Patsali et al. (TAL^A^ and TAL^B^) where C = NN [16]. The minor-case letter for RGN (CPF^A^) gives a deviation from perfect complementarity of the gRNA with the (mutant) target sequence. *Target⇥* and *Exon⇥* give the distance of the predicted cleavage site from the target and from the nearest exon, respectively. Suitable nucleases were first shortlisted by a minimum 5-nt distance of cleavage from exons, then by the distance of the cleavage site from the target site. *Scores* give efficiency and, where available, off-target scores as additional criteria for selection, specifically for Cas9: efficiency score (Doench ‘16)/off-target activity (MIT Specificity Score), Cas12a: efficiency score (DeepCpf1), and TALENs: efficiency score of monomers RVD1/RVD2 (TALEN Finder). The nuclease with the highest efficiency score was selected for inclusion here.

**Table 1 jcm-08-01959-t001:** Shortlisted potentially therapeutic mutation-specific targets for DARE-based functional correction of β-thalassemia. A full list of β-thalassemia mutations passing initial filter criteria for analysis is shown in Appendix A.

IthaID^1^	Common Name	HGVS^2^ Name	Type of Mutation	Region	Exon⇥^3^	References
107	IVS I-5 G>A	HBB:c.92+5G>C	Activation of cSD^4^; partial SD^5^ LoF^6^	SD-proximal	5 nt^7^	[31,32]
111	IVS I-6 T>C	HBB:c.92+6T>C	Activation of cSD; partial SD LoF	SD-proximal	6 nt	[33]
3445	IVSI-13G	HBB:c.92+13	cSD activated by IthaID 101-103^8^, 107, 111	Intron	13 nt	[31]
113	IVS I-110 G>A	HBB:c.93-21G>A	**Confirmed target; GG>GA (aSA^9^)**	aSA	21 nt	[16,17]
3008	IVS I-115 A>T	HBB:c.93-16A>T	AT>TT (effect unclear)	Intron	16 nt	[34]
114	IVS I-116 T>G	HBB:c.93-15T>G	TT>GT (potential aSD^10^)	aSD	15 nt	[35]
208	IVS II-5 G>C	HBB:c.315+5G>C	Activation of cSD; partial LoF	Intron	5 nt	[36,37]
3446	IVS II-579G	HBB:c.316-272	cSA^11^ activated by IthaID 214	Intron	270 nt	[31]
210	IVS II-613 C>T	HBB:c.316-238C>T	Activation of cryptic splice site	Intron	238 nt	[38]
211	IVS II-654 C>T	HBB:c.316-197C>T	**Confirmed target; GC>GT (aSD)**	Intron	197 nt	[17,39]
212	IVS II-705 T>G	HBB:c.316-146T>G	Activation of cryptic splice site	Intron	146 nt	[40,41]
213	IVS II-726 A>G	HBB:c.316-125A>G	Likely block of RNA processing	Intron	125 nt	[42]
214	IVS II-745 C>G	HBB:c.316-106C>G	aSD, activating cSA IthaID 3446	Intron	106 nt	[31]
215	IVS II-761 A>G	HBB:c.316-90A>G	AT>GT (potential aSD)	Intron	90 nt	ITHANET^12^
2183	IVS II-781 C>G	HBB:c.316-70C>G	CT>GT (potential aSD)	Intron	70 nt	[43]
216	IVS II-815 C>T	HBB:c.316-36C>T	CT>GT (potential aSD)	Intron	36 nt	[44]
217	IVS II-837 T>G	HBB:c.316-14T>G	AT>AG (potential aSA)	Intron	14 nt	[45]

^1^ Nucleotide-specific target ID from ITHANET (www.ithanet.eu); ^2^ Human Genome Variation Society; ^3^ Distance of the target from the nearest exon; ^4^ cSD—cryptic splice donor; ^5^ SD—splice donor; ^6^ LoF—loss of function; ^7^ nt—nucleotide; ^8^ IthaIDs 101, 102, and 103 abolish any activity of the normal SD and are unsuitable by DARE-based repair and are not shown; Hbb:c.92+13 is one of three cryptic splice donors activated by these mutations, the other two are exonic and cannot be targeted by DARE; ^9^ aSA—aberrant splice acceptor; ^10^ aSD—aberrant splice donor; ^11^ cSA—cryptic splice acceptor; ^12^ as yet unpublished mutation from www.ithanet.eu.

**Table 2 jcm-08-01959-t002:** Disease categories and exemplary mutations with high relative allele frequency, likely treatable by DARE-based gene therapy.

Primarily Affected	Exemplary	Exemplary Mutations				References
Organ System	Disorders	Gene	dbSNP ID	Effect^1^	Frequency^2^	
Circulatory	Poikilocytic anemia	SPTA1	rs757147440	aSA	25%^3^	[47]
Endocrine	Hyperinsulinemic hypoglycemia	ABCC8	rs151344623	aSA	68.8%^4^	[48,49]
Nervous	Leber congenital amaurosis	CEP290	rs281865192	cSD activation	43%^5^	[50,51,52]
Sensory	Stargardt disease	ABCA4	rs1457937638	cSD activation	7.5%^6^	[53,54]
Muscular	Miyoshi myopathy	DYSF	rs1285082850	cSD activation	17 families^7^	[55]
	Congenital muscular dystrophy	FKTN	rs1554754182	cSD activation	20.8%^8^	[56]
Integumentary	Erythropoietic protoporphyria	FECH	rs2272783	cSA activation	42.6%^9^	[57,58]
Respiratory	Cystic fibrosis	CFTR	rs397508266	aSD	2.0%^10^	[59,60]
Multisystemic	Fabry disease	GLA	rs199473684	cSD activation	41.1%^11^	[61,62]
Cancer	Breast cancer	BRCA2	rs191253965	cSD activation	0.2%^12^	[63]

^1^ Effect of the specific mutation, which might be activation of a cryptic splice donor (cSD) or splice acceptor (cSA), or creation of an aberrant splice donor (aSD) or acceptor (aSA) site. ^2^ Peak relative allele frequency of the mutation for a representative sample, as available and as annotated below for each number given. ^3^ The mutation represented 20 disease alleles in 40 patients. ^4^ The mutation represented 74 in 158 identified disease alleles in patients, with an overall carrier rate of 351 in 21,122 Ashkenazi Jews [48]. ^5^ The mutation represented 33 in 80 identified disease alleles and was present in 28 of 57 patients with identified disease alleles [51]. ^6^ The mutation represented 20 in 261 alleles studies and was represented in 15% (20 of 131) of patients [53]. ^7^ The mutation was detected in 23 patients from 17 independent families [55], and while the study overlaps with a wider dysferlinopathy study of 193 patients [64], neither publication gives the total number of families investigated. ^8^ The mutation was detected in the compound heterozygous condition in five out of 12 patients investigated. ^9^ The mutation was detected in the compound heterozygous condition in 132 out of 155 patients investigated [58]. ^10^ The mutation was detected in 21 chromosomes in 520 affected families [59]. ^11^ The mutation was detected in 37 out of 45 patients, in a screen of 110,027 neonates [61]. ^12^ The mutation was detected in nine out of 2000 families presenting with BRCA1/BRCA2-negative breast and ovarian cancer [63].

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
