# Peer review of "The Scope for Thalassemia Gene Therapy by Disruption of Aberrant Regulatory Elements"

_jcm, 2019, doi:10.3390/jcm8111959_

Round 1

Reviewer 1 Report

This reviewer is curious to know the authors' thoughts on why no mention of ZFNs was made in the introduction and it was not included in the experimental comparison in the work. A bulk part of the gene-editing field has been with ZFNs and the followers, especially CRISPR has seen rapid development as the area had been trail blazed by ZFNs before. I believe that the authors should either clarify their thoughts and reasons on the absence of ZFNs or add in the method in the introduction followed by reasons on why they chose to not included them.

Moreover, this is critical as my understanding is that ZFNs have already seen some success at clinical trials (Hunter/HIV?).....

Minor comment: 

From line 284 onwards, the authors describe and discuss Figure 2 and keep referring to it as Figure 3. Please correct and cross-check.

Reviewer 2 Report

The manuscript "The scope for thalassemia gene therapy by disruption of aberrant regulatory elements" by Patsali et al is an attempt to address a systematic personalized therapeutic approach in the realm of gene editors. The study is a follow-up or branch of the same authors' earlier publication on thalassemia. Given that, the in wet lab and silico approach appeals judicious scientific exploration for genetic disorders.

The manuscript is well written, however, it is highly recommended that a general schematic of workflow or strategy should be included, as the study claims that it can be used for other disease models. This will improve the manuscript impact and lead way for other studies.

It is recommended that the authors submit the sequences of the clones to NCBI or any suitable repository prescribed by the journal and include the accession number in the manuscript. 

Considering the potential and application of this study, acknowledging the limits of the manuscript,  this reviewer is convinced with the findings and would like to recommend the editor for publication. 

Reviewer 3 Report

The report entitled ‘The scope for thalassemia gene therapy by disruption of aberrant regulatory elements’ investigates disruption of aberrant regulatory elements (DARE) as an effective strategy for b-thalassemia gene therapy. Following performing DARE in MEL-HBBIVS cell model, correction of β-globin expression was achieved by either removal of causative mutations and/or through removal of context sequences. Via in silico analyses, the authors conclude that application of DARE may have wide scope for sustainable personalized advanced therapy medicinal product development for thalassemia and other diseases.

Overall, the work is pertinent and the data is interesting. As the majority of the community attempt to optimize the strategy for precisely targeted edits, the authors demonstrate the therapeutic potential of NHEJ-based mutation-specific repair by DARE to a vast number of genetic diseases. This would be of particular importance if the causative mutation itself is not able to be targeted or the precise correction of primary mutations is inefficient. This study may also help to yield a strategy to generate precise indels of defined type/length in future studies.

The article is very well structured and written, making it easy to read. My comments are listed in order as follows:

Pg 7-8, line 284, 286, 314, 317, ‘figure 3’ should read ‘figure 2’. The current format of ‘figure 2, RNA panel’ is confusing. With the relative percentages of aberrant and normal HBB listed between the two bar graphs, at first glance of the data, it is very easy to correlate the number to the bar graph that is next to, however, the height of the column obviously does not match the percentage. As depicted, all bar graphs here are normalized to the mock-edited control (which was assigned the number 1). A couple of suggestions to make it less confusing: Change ‘relative to non-edited clones’ (at the top of the graph) to ‘relative to mock-edited clones’ since there were two unmodified clones (though treated with TALEN) A line through the number 1 to distinguish between increase and decrease of the aberrant transcript expression (with broken x-axis if needed, and clarify that less than 1 indicates decreased expression). Move the list of relative percentages of aberrant vs normal transcript to the left side of the total HBB panel (green bar graph) Please specify that the data was calculated using the delta delta ct method in the figure legend as well.

Below comments are not required to be answered.

There was some very interesting data in figure 2. For example, both clones A1#8 and B1#9 have ­­­the aberrant SA site intact, although clone B1#9 deleted one more base, complete correction of the HBB splicing was achieved in both clones, yet A1 #8 not B1#9 had significantly increased aberrant mRNA level. It seems there is a bias, most of the indel events are upstream deletions, is this cell type (hematopoietic cells) dependent or cut sites dependent or is it related to the nuclease? When applying DARE to primary bulk hematopoietic stem/progenitor cells, the edited cells would be a mix as well. Any thoughts on whether it is necessary and how to manage selection for correct cells in that situation (before transplanting back to patients)?
